Detecting Parkinson’s disease from shoe-mounted accelerometer sensors using convolutional neural networks optimized with modified metaheuristics

Jovanovic Luka 1
http://orcid.org/0000-0001-9990-1084 Damaševičius Robertas 2 robertas.damasevicius@vdu.lt
Matic Rade 3
Kabiljo Milos 3
http://orcid.org/0000-0001-5709-3744 Simic Vladimir 4 5
Kunjadic Goran 6
http://orcid.org/0000-0002-5511-2531 Antonijevic Milos 7
Zivkovic Miodrag 7
http://orcid.org/0000-0002-2062-924X Bacanin Nebojsa 7 8
1 Faculty of Technical Sciences, Singidunum University , Belgrade , Serbia
2 Department of Applied Informatics, Vytautas Magnus University , Akademija , Lithuania
3 Department for Information Systems and Technologies, Belgrade Academy for Business and Arts Applied Studies , Belgrade , Serbia
4 Faculty of Transport and Traffic Engineering, University of Belgrade , Belgrade , Serbia
5 College of Engineering, Department of Industrial Engineering and Management, Yuan Ze University , Taoyuan City , Taiwan
6 Higher Colleges of Technology , Abu Dhabi , United Arab Emirates
7 Faculty of Informatics and Computing, Singidunum University , Belgrade , Serbia
8 MEU Research Unit, Middle East University , Amman , Jordan
Alatas Bilal
Electronic publication date: 2024 May 13
Publication date: 2024
Volume: 10
Electronic Location ID: e2031
Received 2024 Jan 19; Accepted 2024 Apr 9
Copyright: © 2024 Jovanovic et al.
Copyright year: 2024
Copyright holder: Jovanovic et al.
License: This is an open access article distributed under the terms of the Creative Commons Attribution License, which permits unrestricted use, distribution, reproduction and adaptation in any medium and for any purpose provided that it is properly attributed. For attribution, the original author(s), title, publication source (PeerJ Computer Science) and either DOI or URL of the article must be cited.
License URL: https://creativecommons.org/licenses/by/4.0/

Keywords: Parkinson’s disease, Convolutional neural network, Optimization, Extreme gradient boosting, Metaheuristics, Wearable sensors, Smart healthcare

Funding: The authors received no funding for this work.

==============================
Neurodegenerative conditions significantly impact patient quality of life. Many conditions do not have a cure, but with appropriate and timely treatment the advance of the disease could be diminished. However, many patients only seek a diagnosis once the condition progresses to a point at which the quality of life is significantly impacted. Effective non-invasive and readily accessible methods for early diagnosis can considerably enhance the quality of life of patients affected by neurodegenerative conditions. This work explores the potential of convolutional neural networks (CNNs) for patient gain freezing associated with Parkinson’s disease. Sensor data collected from wearable gyroscopes located at the sole of the patient’s shoe record walking patterns. These patterns are further analyzed using convolutional networks to accurately detect abnormal walking patterns. The suggested method is assessed on a public real-world dataset collected from parents affected by Parkinson’s as well as individuals from a control group. To improve the accuracy of the classification, an altered variant of the recent crayfish optimization algorithm is introduced and compared to contemporary optimization metaheuristics. Our findings reveal that the modified algorithm (MSCHO) significantly outperforms other methods in accuracy, demonstrated by low error rates and high Cohen’s Kappa, precision, sensitivity, and F1-measures across three datasets. These results suggest the potential of CNNs, combined with advanced optimization techniques, for early, non-invasive diagnosis of neurodegenerative conditions, offering a path to improve patient quality of life.

Introduction

Neurodegenerative diseases encompass a set of progressive conditions marked by the gradual deterioration and demise of nerve cells (neurons) in either the brain or the peripheral nervous system (Dugger & Dickson, 2017; Kovacs, 2018). Such diseases commonly lead to a decline in cognitive function, movement, and various other neurological functions (Katsuno et al., 2018; Christidi et al., 2018). The most common neurodegenerative diseases include Alzheimer’s disease, Parkinson’s disease, Amyotrophic Lateral Sclerosis, Multiple Sclerosis, Creutzfeldt-Jakob disease, and many others. The precise origins of neurodegenerative diseases are frequently intricate and multifaceted, incorporating genetic, environmental, and lifestyle factors (Hou et al., 2019; Popa-Wagner et al., 2020; Bianchi, Herrera & Laura, 2021). Diagnosis typically entails a blend of clinical assessment, medical history analysis, and, in special cases, imaging or genetic testing (Gómez-Río et al., 2016; Zhou et al., 2020; García & Bustos, 2018; Hansson, 2021). Despite continuous research and progress in comprehending these disorders, numerous neurodegenerative diseases persist. Treatment primarily centers on symptom management and improving the quality of life for persons grappling with these disorders (Maneu, Lax & Cuenca, 2022; Aza et al., 2022; Mortberg, Vallabh & Minikel, 2022).

The diagnostics of neurodegenerative diseases pose several challenges, reflecting the complexity of these conditions and the boundaries of the current medical approaches (Domínguez-Fernández et al., 2023; Kumar et al., 2023). Although early detection is vital for timely intervention and better management of the condition, many disorders, such as Alzheimer’s and Parkinson’s, often manifest symptoms only in the later stages, making early detection challenging (Shusharina et al., 2023; Bhat et al., 2018). Moreover, the overlap of symptoms between different neurodegenerative disorders complicates accurate diagnosis. Distinguishing between conditions with similar clinical presentations is crucial for appropriate treatment and management. Lastly, early symptoms may be very subtle, and many patients seek treatment only when symptoms seriously threaten their quality of life (Morel et al., 2022; Rosqvist, Schrag & Odin, 2022; Meng et al., 2022).

Early diagnosis of Parkinson’s disease (and other neurodegenerative disorders in general) provides several advantages, contributing to better patient outcomes and overall disease management (Stern, 1993; Pagan, 2012; Kobylecki, 2020). It allows the prompt beginning of the treatment, and at the same time medications and therapy can be more efficient in controlling the disease’s symptoms if started early enough, consequently significantly improving the patient’s quality of life (Murman, 2012; Hauser et al., 2009; Armstrong & Okun, 2020). From the healthcare point of view, medical staff can monitor the progression of the disease over time, allowing tailored treatment plans to the individual patients, minimizing the symptoms’ impact on patient’s daily activities (van Halteren et al., 2020; Fröhlich et al., 2022). Reduced economic burden should not be neglected as well, as early diagnostics can minimize the necessary hospitalizations and visits to the emergency rooms. Therefore, it allows more efficient medical staff and resource allocation that will in the long run reduce the overall costs (Yang et al., 2020; Boina, 2022; Soh et al., 2022; Radder et al., 2020).

Data-driven models for Parkinson’s disease detection leverage different types of sensors and contemporary technologies to collect and analyze data related to movement, walking patterns (Priya et al., 2021), handwriting tremors (Bernardo et al., 2022), phonation (Hashim et al., 2023), computer keypress time data (Bernardo et al., 2021) and other symptoms commonly associated with Parkinson’s disorder (Khoury et al., 2019; Dai, Tang & Wang, 2019; Chandrabhatla, Pomeraniec & Ksendzovsky, 2022). These relatively cheap wearable sensors play a crucial role in remote monitoring, early detection, and ongoing management of Parkinson’s disease. Wearable gadgets, like smartwatches and fitness trackers, frequently incorporate accelerometers and gyroscopes, enabling them to track patterns of movement and identify subtle changes in gait and tremor (Mughal et al., 2022; Reichmann, Klingelhoefer & Bendig, 2023; Schalkamp et al., 2023). Gait-freezing, for instance, is a typical symptom observed in individuals with Parkinson’s disease, defined as a sudden and temporary inability to initiate or continue walking despite the willingness to move (Pozzi et al., 2019; Gao et al., 2020; Lewis et al., 2022; Moreira-Neto et al., 2022). This phenomenon is easily detectable by a simple gyroscopic system built in the patient’s shoes (Pardoel et al., 2019; Marcante et al., 2021).

Methods belonging to artificial intelligence (AI) and subcategories like machine learning (ML) are pivotal in the early identification and surveillance of Parkinson’s disease, employing diverse techniques such as machine learning and data analysis (Senturk, 2020; Belić et al., 2019; Gupta et al., 2023). These methodologies analyze intricate datasets to discern patterns that signify the presence of the disease. For example, AI methods are capable of analyzing medical imaging data, such as magnetic resonance imaging (MRI) or computed tomography (CT) scans, to discover subtle changes in brain structures associated with Parkinson’s diseases (Xu & Zhang, 2019; Zhang, 2022; Francis, Rajan & Pandian, 2022). Additionally, AI approaches are capable of analyzing data from wearable devices to track changes in motor function, such as gait abnormalities or tremors linked to conditions like Parkinson’s disease (Balaji, Brindha & Balakrishnan, 2020; Wu et al., 2023). These algorithms may also aid in examining genetic and biomarker data to pinpoint specific markers related to particular diseases. Consequently, AI methods can help differentiate among various neurodegenerative diseases that might exhibit similar clinical symptoms but possess distinct underlying pathology (Khoury et al., 2019; Thapa et al., 2020; Noor et al., 2019). In this way, doctors are enabled to make better-informed decisions.

Recognizing the potential benefits of data-driven diagnostic approaches is vital, nevertheless, it is important to acknowledge the specific challenges they pose. Sufficient volumes of data are required, and the quality and consistency of the data are paramount. Ethical considerations concerning data collection and processing, patient consent, and data privacy require thorough attention. Moreover, it is vital to verify and regularly enhance AI models to ensure their accuracy, making them suitable for clinical use. Nonetheless, AI techniques show great promise, as evident in recent publications (Singh et al., 2019; Tăuţan, Ionescu & Santarnecchi, 2021; Lin et al., 2020; Khaliq et al., 2023). Another advantage of employing AI approaches is the possibility to apply feature analysis tools like Shapley Additive Explanations (SHAP) (Lundberg et al., 2020), which can contribute to a better comprehending of both the particular disorder and diagnostic process itself (Liu et al., 2022; McFall et al., 2023; Junaid et al., 2023). SHAP analysis improves the transparency of the model, interpretability, and trust in the obtained results, therefore allowing an informed decision-making process in general. Interpretability of the outcomes is vital, as it provides clear insight into each feature’s contribution to the overall model forecasts. Feature importance, on the other hand, allows quantification of each feature’s significance, therefore enabling feature prioritization with respect to their impact on the model’s forecasts.

The central difficulty in the domain of AI and ML is centered on identifying the suitable values for the hyperparameters of the model being used. This problem is accentuated by the principle encapsulated in the “no free lunch” theorem (NFL) (Wolpert & Macready, 1997), underlying that there is no universally superior method for consistently outperforming all others across a diverse range of problems. Essentially, this theorem underscores the need to tailor hyperparameter configurations for each unique problem to attain satisfactory performance. Failing to choose the optimal hyperparameters inevitably leads to a suboptimal level of performance of the utilized model. Manual fine-tuning of a model for each specific problem is an exceptionally intricate and time-intensive procedure, which is inherently an NP-hard optimization challenge. Therefore, conventional deterministic algorithms are not appropriate for resolving it. In the domain of stochastic methods, metaheuristics approaches are regarded as very powerful optimization tools, exhibiting considerable potential in this field, which is evidenced by a significant number of recent relevant publications (Todorovic et al., 2023; Petrovic et al., 2023; Zivkovic et al., 2023; Bacanin et al., 2023; Nematzadeh et al., 2022; Esmaeili, Bidgoli & Hakami, 2022; Chou et al., 2022; Abbas et al., 2023; Chou, Nguyen & Chang, 2022).

This manuscript addresses the analysis of gait, a critical aspect in the process of diagnosing Parkinson’s disease (Jankovic, 2015; Von Coelln et al., 2021; Mirelman et al., 2019). Recent relevant studies have emphasized the significance of gait analysis (Wang et al., 2023), revealing that perturbations in gait can manifest during the early stages of the disease (Pistacchi et al., 2017; Di Biase et al., 2020; Ghislieri et al., 2021). Common characteristics of gait anomalies in Parkinson’s disorder include shuffling steps, diminished arm swinging, freezing during walking, and postural instability (Perumal & Sankar, 2016; Morris et al., 2001). These anomalies are associated with the underlying drop of dopamine and other alterations in brain activity affecting motor control and coordination. Considering that alterations in gait represent among the earliest symptoms of the disorder, a robust gait classification can prove highly valuable for medical staff, aiding them in the diagnostic process.

A research gap is present in the literature in terms of observing an immense amount of collected sensor data in the form of images to better address positional relations in the data while reducing computational demands through limited local connectivity in CNN. The use of CNNs is well established throughout the fields of computer vision, where CNNs excel in classification tasks. However, their application is challenging when observing continuous data measurements, such as the data acquired from the shoe-mounted sensors. By converting the sensor data to the image format, it can be conveniently used as an input for a CNN classifier. Therefore this research seeks to address this literature gap utilizing a convolutional neural network (CNN) to reduce the number of attributes, as the employed real-world medical dataset is intricate. This approach has yet to be explored in the literature and has yet to be applied to the detection of Parkinson’s disease. XGBoost model is then used to produce the final classification. Additionally, an improved version of the novel sinh cosh optimizer (SCHO) (Bai et al., 2023) has been employed to optimize the hyperparameters of the model for this specific challenge. As one of the most recent additions to the metaheuristics family, the potential of SCHO has not yet been thoroughly explored. Moreover, the empirical trials that were executed before the main experiments have shown that the elementary version of SCHO attains very promising results, and it was consequently chosen for further improvements. Hence, the principal contributions of this research can be succinctly outlined as follows: An improved version of the SCHO algorithm was devised, to enhance the elementary variant of the metaheuristics.

This devised algorithm was incorporated as the component of the ML framework to discover the optimal collection of hyperparameters for the particular gait analysis problem.

The assessment of the proposed model was conducted with the standard gait dataset which is associated with Parkinson’s disorder. The simulation results were subsequently juxtaposed with the models optimized by alternative state-of-the-art metaheuristics algorithms, accompanied by a statistical assessment of the simulation results.

Exploring a unique perspective on Parkinson’s sensor data in the form of images in combination with CNN to better determine relations between data points.

SHAP has been employed to interpret the obtained outcomes, providing a deeper comprehension of both the model and the significance of the attributes.

The rest of the manuscript is prepared as follows. In “Background”, a literature survey on AI and data-driven diagnostic procedures is conducted, as well as medical classification problems. Additionally, a brief overview of the CNNs, XGBoost model, and metaheuristics optimization is given. “Methods” initially presents the plain version of SCHO metaheuristics, outlines its limitations, and suggests alterations to improve the algorithm. The experimental setup is detailed in “Setup”, while “Results” encompasses the simulation outcomes, and statistical assessment, followed by the SHAP analysis of top-performing models. Concluding remarks and directions for future research activities in this challenging field are presented in “Conclusion”.

Background

The incorporation of AI and data-driven methodologies presents numerous advantages in medicine, particularly in diagnostic procedures (Dai, Tang & Wang, 2019; Anikwe et al., 2022; Basile et al., 2023; Lee & Yoon, 2021). Modern diagnostic methods within Healthcare 4.0, encompassing the integration of Internet of Things (IoT) gadgets, generate a substantial data influx, which is observed as a trend that continues to escalate (Krishnamoorthy, Dua & Gupta, 2023; Kishor & Chakraborty, 2022; Greco et al., 2020; Javaid & Khan, 2021). AI methods demonstrate the capability to rapidly and accurately assess complex datasets, and in many cases surpass human medical experts in the diagnostics of different diseases and disorders. These models excel in identifying delicate patterns and disparities that humans may evade. Moreover, AI approaches exhibit high efficiency, supporting early diagnostics by discovering subtle illness markers in their earliest stages, and facilitating prompt interventions and correct choice of treatment (Hunter, Hindocha & Lee, 2022; Paul et al., 2022; Rashid et al., 2022; Van der Schaar et al., 2021). Early detection is associated with enhanced chances of patient recovery and a significant decrease in overall healthcare costs (Johnson, Albizri & Simsek, 2022; Rajpurkar et al., 2022; Muhammad et al., 2020).

The application of AI streamlines the decision-making process for healthcare experts, supporting prompt diagnostics and decreasing patient waiting delays, consequently improving the general efficacy of the healthcare systems (Basile et al., 2023; Tang et al., 2021; Stewart et al., 2023; Lång et al., 2023; Alowais et al., 2023). Well-developed trailblazing AI frameworks consistently produce results, regardless of the time of day or medical provider’s practical knowledge, contributing to a reduction in errors attributed to human factor (Yeasmin, 2019; Haleem et al., 2020; Gaba, 2018). Furthermore, AI-powered diagnostics may generate significant cost reduction through optimization of resource allocation, minimization of unnecessary tests, and prevention of wrong or late diagnoses (Blasiak, Khong & Kee, 2020; Munavalli et al., 2021; Lång et al., 2021; Dembrower et al., 2020).

Despite numerous publications dealing with the growing interest and potential of AI in medicine, and without a doubt numerous positive facets, some common drawbacks must be considered critically. First of all, claiming that AI can outperform human doctors and revolutionize medicine overnight leads to overambitious expectations, and finishes with undermined trust if the concepts are not implemented fast enough. Another point to be highlighted here is the shortage of interpretability and transparency. It is vital to comprehend why some model has made a particular prediction, notably in the medical domain, where each decision made may have life-changing consequences for the patient. Lastly, the implementation of these models in practice demands considerable resources, like funds, infrastructural changes, and training for medical workers.

Convolutional neural networks

Convolutional neural networks (CNNs or ConvNets) represent a class of deep neural networks specifically crafted for tasks related to visual data, such as images and videos. Renowned for their exceptional efficacy in computer vision applications, CNNs have consistently demonstrated cutting-edge performance across diverse challenges, like image classification, object detection, segmentation, and beyond (Li et al., 2021; Gu et al., 2018; Yamashita et al., 2018).

The core of the CNNs consists of the convolutional layers, which perform convolution operations over input data by applying filters (kernels) to identify patterns and features. Afterward, pooling layers reduce the spatial dimensions while retaining important features. Activation functions, like ReLU, are used for introducing the non-linear element to the network, which allows it to acquire intricate relations within the data. Fully connected layers, connecting each neuron in one layer to every neuron in the following layer, are employed in the final stages to perform classification tasks. Flattening is utilized to convert the output of the convolutional and pooling layers, which is fed to the fully connected layers. Batch normalization is commonly applied to standardize the input of every layer, aiding in the stabilization and acceleration of the training process. Finally, dropout serves as a regularization method wherein randomly selected cells are removed during the training phase to mitigate the risk of overfitting.

The accuracy of the model is heavily influenced by hyperparameters, making them a crucial aspect of optimization (Wang, Zhang & Zhang, 2019). Examples of hyperparameters include the number of kernels and their size in each convolutional layer, the learning rate, batch size, the architecture involving the count of convolutional and fully-connected (dense) layers, weight regularization in dense layers, the choice of activation function, the dropout rate, and more. Hyperparameter optimization is not a universally solvable process for all problems, necessitating a “trial and error” approach. However, these approaches are time-consuming and offer no guarantee of outcomes, contributing to their classification as NP-hard. Metaheuristics algorithms have shown promising outcomes in handling such challenges (Yamasaki, Honma & Aizawa, 2017; Qolomany et al., 2017; Bochinski, Senst & Sikora, 2017). For an in-depth mathematical formulation of CNN, refer to Albawi, Mohammed & Al-Zawi (2017), and a more recent exploration on the same topic is provided in Gu et al. (2018).

CNNs are frequently used for image and video recognition, medical image analysis, face recognition, and more (Krizhevsky, Sutskever & Hinton, 2012; Ranjan et al., 2017; Balaban, 2015; Spetlík, Franc & Matas, 2018; Cai, Gao & Zhao, 2020; Ting, Tan & Sim, 2019). A particularly important application of CNNs is the field of medical images, where they are successfully applied for the classification of brain tumors (Bíngol & Alatas, 2021; Bezdan et al., 2021b), breast cancer (Zuluaga-Gomez et al., 2021) and thoracic diseases (Abiyev & Ma’aitaH, 2018). Renowned pre-trained CNN architectures such as AlexNet (Krizhevsky, Sutskever & Hinton, 2017), VGGNet (Simonyan & Zisserman, 2014), ResNet (Szegedy et al., 2017), Inception (Soria Poma, Riba & Sappa, 2020), and MobileNet (Wang et al., 2020) have seen broad adoption across diverse tasks. It is noteworthy that, although CNNs are predominantly linked with computer vision tasks, their application extends beyond. They have been successfully employed in processing other types of data, such as one-dimensional signals in speech and audio processing.

XGBoost

The XGBoost (Chen & Guestrin, 2016) technique leverages a decision tree-based ensemble learning strategy to combine forecasts from numerous weak learners. Each tree, using a gradient-boosting framework, corrects faults caused by its ancestors. The efficacy of XGBoost is based on its regularization methods and parallel processing efficiency. Aside from optimization, regularization and gradient boosting can improve performance. The XGBoost model predicts using intricate relationships between input and target patterns. To improve the objective function, the XGBoost method employs an incremental training strategy. Due to an immense count of parameters requiring adjustment when tuning XGBoost, the trial and error technique is impractical. Given the intricacy of some situations, a strong model is required. The primary characteristics of a good model are speed, generalization, and accuracy.

To produce the best outcomes, the model should be trained iteratively. The objective function of the XGBoost is described by the Eq. (1)

(1) obj(Θ)=L(θ)+Ω(Θ),

where Theta is the collection of XGBoost hyperparameters, L(Theta) is the loss function, and Omega(Theta) is the regularization term. The last parameter controls the model’s complexity. The loss function depends on the specific problem being addressed.

(2) L(Θ)=∑i(yi−y^i)2,

in which the yi is the predicted value, while the predicted target for each iteration i is y^i.

(3) L(Θ) = ∑i[ yiln⁡ (1+e−y^i) + (1−yi)ln⁡ (1+ey^i)].

The purpose of this procedure is to distinguish between real and expected values. The total loss function is minimized to enhance classification.

Metaheuristics optimization

The realm of metaheuristics optimizers gained popularity due to their efficiency in resolving NP-hard tasks. The biggest hurdle is discovering solutions within an acceptable time frame and upholding manageable hardware demands. These methods may be separated into distinctive groups, but there is no strict definition. One classification commonly acknowledged by the majority of scientists is distinguishing them concerning the phenomena that inspire these algorithms. Thus, these distinctive families comprise light, swarm, genetic, physics, human, and the most recent addition in the form of mathematically inspired approaches. For example light-based methods are inspired by light propagation properties (Alatas & Bingol, 2020), and the notable techniques include ray optimization algorithm (RO) (Kaveh & Khayatazad, 2012) and optics-inspired optimization (OIO) (Kashan, 2015; Bingol & Alatas, 2020).

Drawing motivation from breeds that thrive in huge swarms and benefit from collective behavior, swarm-inspired algorithms are particularly effective when a sole individual is insufficient to accomplish a task. The swarm family of algorithms was established as highly effective in solving NP-hard problems, but to optimize their performance, it is recommended to hybridize them with similar algorithms. The challenge with these stochastic population-based methods lies in their tendency to favor either exploration or exploitation. This can be addressed by incorporating mechanisms from different solutions. Notable approaches include particle swarm optimization (PSO) (Eberhart & Kennedy, 1995), genetic algorithm (GA) (Mirjalili & Mirjalili, 2019), sine cosine algorithm (SCA) (Mirjalili, 2016), firefly algorithm (FA) (Yang & Slowik, 2020), grey wolf optimizer (GWO) (Faris et al., 2018), reptile search algorithm (RSA) (Abualigah et al., 2022), red fox algorithm (Połap & Woźniak, 2021), polar bear algorithm (Polap & Woźniak, 2017), and the COLSHADE algorithm (Gurrola-Ramos, Hernàndez-Aguirre & Dalmau-Cedeño, 2020).

Swarm algorithms find practical applications across a diverse array of real-world challenges. These applications span various domains, including glioma MRI classification (Bezdan et al., 2020), credit card fraud detection (Jovanovic et al., 2022a; Petrovic et al., 2022), global optimization problems (Strumberger et al., 2019; Zamani, Nadimi-Shahraki & Gandomi, 2022; Nadimi-Shahraki & Zamani, 2022). Additionally, swarm metaheuristics are successfully employed in cloud computing (Predić et al., 2023; Bacanin et al., 2019), enhancing the audit opinion forecasting (Todorovic et al., 2023) predicting the number of COVID-19 cases (Zivkovic et al., 2021), software engineering (Zivkovic et al., 2023), feature selection (Bezdan et al., 2021a; Jovanovic et al., 2022b; Stankovic et al., 2022), security and intrusion detection (Savanović et al., 2023; Jovanovic et al., 2022c; Salb et al., 2023), and enhancing wireless sensor networks (Zivkovic et al., 2020a, 2020b).

The authors in Ahmadpour, Ghadiri & Hajian (2021) devised a genetic algorithm-grounded method for monitoring patients’ blood pressure, leading to a significant enhancement in their overall quality of life and enabling the early diagnosis of some preventable illnesses. Khan & Algarni (2020) investigated an IoT environment that facilitates continuous monitoring of patients’ conditions, resulting in substantial improvements in their cardiovascular health. Illustrative instances of AI-assisted medical diagnosis encompass the detection of diabetic retinopathy (Gupta & Chhikara, 2018), classification of skin lesions (Mahbod et al., 2019), categorization of lung cancer (Ren, Zhang & Wang, 2022), and applications such as magnetic resonance imaging (MRI) and X-ray imaging within the medical field (Zivkovic et al., 2022; Budimirovic et al., 2022).

Materials and Methods

This section begins by presenting the basic sinh cosh optimizer, followed by the suggested modifications that improve the performance of the original implementation.

The original sinh cosh optimizer

SCHO is a recent metaheuristics algorithm developed by Bai et al. (2023). It is a mathematically inspired method, as it relies on the properties of sinh and cosh. Hyperbolic functions encompass common trigonometric functions, with sinh and cosh being fundamental examples. Metaheuristic algorithms may benefit from two key characteristics of cosh and sinh. Firstly, cosh values consistently exceed one, serving as a crucial threshold between exploration and exploitation. Secondly, sinh values fall within the interval [−1, 1], approaching zero, thereby enhancing both exploration and exploitation aspects.

As a metaheuristic relying on population-based methods, the algorithm sets an initial population characterized by a considerable degree of randomness, as illustrated in Eq. (4).

(4) A=[a1,1...a1,j...a1,Da1,2...a2,j...a2,DaN,1...aN,j...aN,D]

In this context, P represents a group of solutions, where the position Ai,j of each agent is calculated according to Eq. (5). Here, the variables D and N signify the dimensional space of the solution and the number of solutions, respectively.

(5) a=rnd(N,D)×(ub,lb)+lb

where rnd represents an arbitrary value, while ub and lb mark the upper and lower boundaries of the search domain.

After the initialization phase, the algorithm must strike a balance between exploration and exploitation, directing solutions toward promising regions within the search space. Exploration is divided into two strategies, and the equilibrium is controlled by Eq. (6):

(6) S=floor(Tct)

here T represents the maximum count of rounds, and ct denotes a control parameter with value empirically determined to be 3.6.

In the exploration phase, solutions are updated as defined by Eq. (7):

(7) A(i,j)t+1={Abest(j)+r1×W1×A(i,j)tr2>0.5Abest(j)−r1×W1×A(i,j)tr2∖lt0.5

In this context, t represents the iteration number, At+1(i,j) describes the j-th dimension of the i-th agent, and A(j)best denotes the best agent in the j dimension. Random values within the range of [0,1] are chosen for r1 and r2. The coefficient W1 signifies a weighted coefficient for the specific agent and can be calculated as follows:

(8) W1=r3×b1×(coshr4+μ×sinhr4−1)

The value of b1 is progressively reduced throughout the iterations, while r3 and r4 are randomly chosen within limits [0,1]. Additionally, a sensitivity parameter is defined and denoted as μ.

During exploration, the second strategy involves the application of Eq. (9)

(9) A(i,j)t+1={Abest(j)+|ϵ×W2×Abest(j)−Ai,j(t)|r5>0.5Abest(j)−|ϵ×W2×Abest(j)−Ai,j(t)|r5<0.5

Within this equation, ϵ is configured to the recommended value of 0.003 from the original publication. The weight coefficient W2 is calculated as follows:

(10) W2=r6×b2

where r6 represents a random number drawn from [0,1], while b2 represents a slowly decreasing value.

Another significant phase of the optimization process is exploitation, when solutions concentrate on promising regions of the search realm, taking more refined steps in the direction of the optima. One more time, the metaheuristic employs two techniques. The initial stage utilizes Eq. (11).

(11) A(i,j)t+1={Abest(j)+r7×W3×A(i,j)tr8>0.5Abest(j)−r7×W3×A(i,j)tr8∖lt0.5

the parameters r7 and r8 are selected within limits [0,1], while W3 is established as follows:

(12) W3=r9×b1×(coshr10+μ×sinhr10)

where r9 and r10 denote arbitrarily chosen values within [0,1].

The second technique relies on the Eq. (13):

(13) A(i,j)t+1=A(i,j)t+r11×singr12coshr12|W2×Abestt−Ai,jt|

where r11 and r12 are arbitrarily chosen within [0,1].

The modified SCHO algorithm

While the basic SCHO algorithm performs well, being a relatively new method, it has plenty of space for improvement. Exploration inside this algorithm is lacking, according to evaluation utilizing CEC (Jiang et al., 2018) standard assessment methodologies. The improved version seeks to address this restriction by incorporating two more strategies.

The first technique is based on the artificial bee colony (ABC) (Karaboga & Basturk, 2008) algorithm. When tired solutions fail to improve, they are discarded and replaced with newly created solutions. Because there are only two iterations in this experiment, solutions that fail to demonstrate improvement are discarded after two iterations if no progress is seen. This method has been shown to improve exploration.

The second approach mentioned is quasi-reflective learning (QRL) (Fan, Chen & Xia, 2020), which is used to generate unique solutions and advance research. This approach is also used to produce potential solutions during the algorithm’s early stages. A given solution X’s quasi-reflected counterpart z is calculated as follows:

(14) Xzqr=rand(lbz+ubz2,xz)

here lb and ub represent the lower and upper limitations of the search realm and rand marks an arbitrary value inside the observed interval. The suggested method is labeled simply modified SCHO (MSCHO). The pseudocode of the proposed optimizer is exhibited in Algorithm 1.

Algorithm 1 Pseudocode of the suggested MSCHO algorithm.

 1:  Set initial parameter values	
 2:  Initialize agent populace P by applying QRL	
 3:  while T > t do	
 4:    Utilized appropriate SCHA search technique to reposition agents	
 5:    Compute agent objective function outcome	
 6:    for each agent A in P do	
 7:      if A has not shown improvement in 2 iterations then	
 8:        Initialize a new agent a by applying QRL	
 9:      end if	
10:    end for	
11:  end while	
12:  return Best attained agent in P	

Introduced framework

In this work, a two-layer framework is introduced to reduce computational demands while still maintaining a holistic observation of patient gait. Each patient’s walking gait recordings consist of several sensor readings over time. It can be difficult to observe this data using a simple approach without the use of feature engineering. This can in turn reduce the amount of information available to the algorithm when making a decision. An additional drawback of this approach is that different studies may require the application of different techniques depending on the sensors used and procedures followed during the study.

A flowchart of the framework can be observed in Fig. 1. The framework is capable of automatically adapting to the given task. The two layers in the framework form a collaborative unit that seeks to improve overall accuracy through feature selection and parameter optimization. The first layer of the framework leverages a CNN for feature selection. The second layer of the framework applies the XGBoost algorithm optimized by several contemporary metastatic tasked with demonstrating the best objective function outcomes.

Figure 1 Flowchart of the introduced framework.

The CNN parameters of the first layer of the framework are optimized by the introduced MSCHO algorithm. Respective optimization ranges empirically determined and for this experiment are as follows; learning rate [0.01,0.0001], count of training epochs [5,25], count of convolutions layers [1,3], units in each the convolutional layer [8,64] number of output features [8,16]. Models are tuned and trained until a predetermined accuracy is surpassed. In this work, an accuracy threshold of 90% is used. The final constructed model architecture is comprised of one convolutional layer with 16 neurons. A learning rate of 0.001225, 20 training epochs are selected by the MSCHO algorithm to provide the best outcomes. Pooling layers are not optimized in this study. The optimization aims to attain a lighter CNN architecture suitable for feature selection while maintaining low resource demands. A depiction of the best architecture is provided in Fig. 2.

Figure 2 Framework layer 1 CNN model architecture.

The reduced feature set is used by the second layer of the framework to further enhance performance. An interpretation of the best constructed CNN utilizing the SHAP (Lundberg & Lee, 2017) model on a selection of 10 random samples is provided in Fig. 3. Input images were zoomed in to improve readability. Once a reduced feature space is determined several metaheuristics are tasked with constructive well-tuned XGBoost models. The specifics of the setup for XGBoost are provided in detail in the following section.

Figure 3 Best constructed CNN model feature importance according to SHAP explainer.

Experimental setup

To assess the introduced approach for Parkinson’s detection from patient gait data from publicly available sources is used (https://physionet.org/content/gaitpdb/1.0.0/). The utilized dataset encompasses three clinical studies (Hausdorff et al., 2007; Frenkel-Toledo et al., 2005; Yogev et al., 2005). Gait pattern measurements for 93 patients with a clinical diagnosis of Parkinson’s are present in the dataset. Additionally, gait measurements for 73 healthy individuals are included in the control group. Patient gait data is collected using a network of 16 sensors positioned on the patient’s shoe soles as depicted in Fig. 4.

Figure 4 Shoe sole sensor positions.

Each sensor output is digitized at a sampling rate of 100 Hz. An additional two columns are conducted in the dataset that represents the sum of all the sensors for each foot. This is done to account for variance in patient weight and variance in balance. More precisely, these columns account for patient weight balance on each foot, as different patients have different weights, therefore the intensity they exert on accelerometers (primarily on impact with the floor) will be slightly different. Additionally, the patients do not have the same center of mass, as they gravitate toward a slightly different balance point in their gait (due to injury, poor posture, and other factors). Therefore, the sum of all values on a given foot at a given moment is added together and introduced as a final column. Using this data, it is possible to examine the force record in relation to time and location, generate metrics that represent the center of pressure over time, and establish timing metrics for each foot.

To enable efficient processing of the available data via CNN-s, patient gait recordings are converted into a 2-dimensional black and white image. It has a similar format to the image shown in Fig. 5. However, the figure shown in the manuscript is resized and stretched to improve readability. The images used for the dataset have significantly lower sizes approximately 20 by 100 pixels with a small variance depending on patient pace.

Figure 5 Sample generated gait image.

Each patient’s gait recordings are separated into batches of 20 steps per sample and appropriately labeled as per patient diagnosis. Each dataset is then separated into training and testing sections. Of the available data 70% is used to train and optimize models, and the remaining 30% is withdrawn for testing. Three experiments are conducted with each of the available datasets.

Simulations in the second layer of the framework involve applying several metaheuristics to the optimization of XGBoost hyperparameters. The optimized hyperparameters are selected due to their high influence on model performance and their respective ranges include the learning rate [0.1,0.9], minimum child weight [1,10], subsample [0.1,1.0], colsample by tree [0.01,1.00], max depth [3,10] and γ [0,0.8]. Respective ranges have been empirically determined.

Several algorithms are included in a comparative analysis to determine the effectiveness of the introduced approach. These include the original SCHO (Bai et al., 2023) algorithm, SCA (Mirjalili, 2016), GA (Mirjalili & Mirjalili, 2019), PSO (Eberhart & Kennedy, 1995), FA (Yang & Slowik, 2020), WOA (Mirjalili & Lewis, 2016), BSO (Shi, 2011), RSA (Abualigah et al., 2022), COA (Jia et al., 2023) and COLSHADE (Gurrola-Ramos, Hernàndez-Aguirre & Dalmau-Cedeño, 2020) algorithms. All metaheuristics are implemented under identical conditions using parameters suggested in the works that originally introduced the algorithm. The population size of 10 agents is used by each algorithm with 15 iterations allocated to improve outcomes. To account for the randomness inherent in metaheuristics algorithms simulations are carried out through 30 experiments. The restricted number of iterations was selected due to the high computational requirements of the simulations. Nevertheless, during empirical trials, it was observed that all algorithms converged successfully within 15 iterations, and additional rounds would not yield significant improvements. In addition to optimized models, a baseline XGBoost implementation without optimization is included in the comparison.

As the evaluated algorithms optimized classification models, several metrics are included in the assessment to ensure a thorough evaluation. Apart from the standard accuracy, precision, sensitivity, and F1-measure metrics shown in Eqs. (15)–(18), the Cohen’s kappa (Warrens, 2015) metric is also tracked during experimentation calculated according to Eq. (19). These metrics are crucial in the evaluation of the model’s performance level. For example, precision is very useful in the cases of imbalanced datasets and where the cost of false positives is high, as it focuses solely on the positively classified entries. On the other hand, sensitivity is vital in scenarios where it is important to correctly identify positive entries, where high values of sensitivity indicate that the model is efficient in capturing the majority of relevant instances. Based on precision and sensitivity together, the F1-measure is utilized to establish the classification threshold. The larger values of the F1-measure suggest that the regarded model has an appropriate balance betwixt precision and sensitivity, thus being capable of efficiently classifying entries belonging to both classes, without favoring one over another. Finally, Cohen’s kappa value measures the inter-rater agreement between classifiers, and it is particularly useful when dealing with imbalanced datasets, as it is considered to be a very robust and reliable indicator.

(15) Accuracy=TP+TNTP+FP+TN+FN

(16) Precision=TPTP+FP

(17) Sensitivity=TPTP+FN

(18) F−1score=2⋅Precision⋅SensitivityPrecision+Sensitivity

here the TP, TN, FP, and FN, denote true positive, true negative, false positive, and false negative values respectively.

(19) κ=po−pe1−pe=1−1−po1−pe

where po represents an observed value while pe is the expected. By utilizing Cohen’s kappa score the optimization challenge is stated as a maximization task.

Simulation outcomes

Simulations are carried out through three experiments, each with one of the available datasets. In each simulation, metaheuristics are tasked with accurately classifying patient gait. Following the simulations, the attained scores are methodically statistically verified.

Yogev et al. (2005) dataset simulations

Comparisons in terms of best, worst mean and median executions for the objective function are shown in Table 1 and in terms of indicator (Cohen’s kappa) are similarly shown in Table 2. Stability comparisons are shown in terms of Std and Var.

Table 1 Yogev et al. (2005) dataset objective function outcomes.

Method	Best	Worst	Mean	Median	Std	Var	
CNN-XG-MSCHO	0.029499	0.032448	0.030973	0.030973	0.001475	2.18E−06	
CNN-XG-SCHO	0.032448	0.038348	0.037365	0.038348	0.002199	4.83E−06	
CNN-XG-SCA	0.029499	0.032448	0.030482	0.029499	0.001391	1.93E−06	
CNN-XG-GA	0.029499	0.038348	0.032448	0.030973	0.003406	1.16E−05	
CNN-XG-PSO	0.029499	0.038348	0.033432	0.032448	0.003679	1.35E−05	
CNN-XG-FA	0.029499	0.035398	0.031465	0.030973	0.002199	4.83E−06	
CNN-XG-WOA	0.029499	0.032448	0.030973	0.030973	0.001475	2.18E−06	
CNN-XG-BSO	0.032448	0.035398	0.032940	0.032448	0.001099	1.21E−06	
CNN-XG-RSA	0.029499	0.035398	0.032940	0.033923	0.002648	7.01E−06	
CNN-XG-COA	0.029499	0.038348	0.034415	0.035398	0.002781	7.73E−06	
CNN-XG-COLSHADE	0.032448	0.035398	0.034415	0.035398	0.001391	1.93E−06	

Table 2 Yogev et al. (2005) dataset indicator (Cohen’s kappa) outcomes.

Method	Best	Worst	Mean	Median	Std	Var	
CNN-XG-MSCHO	0.934485	0.927779	0.931132	0.931132	0.003353	1.12E−05	
CNN-XG-SCHO	0.927779	0.914279	0.916467	0.914279	0.005061	2.56E−05	
CNN-XG-SCA	0.934485	0.928088	0.932301	0.934485	0.003090	9.55E−06	
CNN-XG-GA	0.934485	0.914279	0.927651	0.930976	0.007824	6.12E−05	
CNN-XG-PSO	0.934485	0.914279	0.925353	0.927623	0.008423	7.10E−05	
CNN-XG-FA	0.934485	0.921043	0.930061	0.931287	0.004986	2.49E−05	
CNN-XG-WOA	0.934485	0.928088	0.931287	0.931287	0.003199	1.02E−05	
CNN-XG-BSO	0.927779	0.920702	0.926547	0.927779	0.002617	6.85E−06	
CNN-XG-RSA	0.934485	0.921043	0.926703	0.924580	0.005981	3.58E−05	
CNN-XG-COA	0.934485	0.914279	0.923273	0.921043	0.006408	4.11E−05	
CNN-XG-COLSHADE	0.927779	0.921043	0.923405	0.921550	0.002992	8.95E−06	

In Table 1, CNN-XG-MSCHO exhibits the lowest Best value, indicating strong potential in optimal scenarios. CNN-XG-BSO stands out for its consistency, having the lowest standard deviation and variance, suggesting predictable performance.

In Table 2, the CNN-XG-MSCHO, CNN-XG-SCA, and CNN-XG-WOA methods consistently show high Best, Mean, and Median Kappa values, suggesting they are highly effective. The CNN-XG-GA and CNN-XG-PSO methods have wider ranges in performance (noted in their higher standard deviation and variance), indicating less consistent but potentially high-quality outcomes. This suggests a trade-off between achieving peak performance and maintaining consistency across different evaluations.

A graphical comparison for the objective and indicator functions outcome distributions is provided in Fig. 6.

Figure 6 Yogev et al. (2005) objective and Cohen kappa distributions plots.

Distribution plots in Fig. 6 indicate that a high stability is demonstrated by the introduced algorithm in terms of objective and indicator function outcomes suggesting a strong reliability of the tested algorithms. An interesting observation can be made on the violin plots of the objective function. Multiple peaks in the objective function suggest that two local minimums might be present in the optimization space for this problem. However the SCA algorithm, as well as the introduced modified version managed to overcome this local optima attaining better overall outcomes, something the original SCHO algorithm did not manage to overcome. The improvements introduced by the modified mechanisms are therefore evident. A strong convergence hindered several algorithms in the simulation. Algorithms such as the original SCHO, BSOA, and COLSHADE showcased a limited exploration ability. Despite the BSO algorithm attaining the highest stability it also fails to locate the best solution within the search space relative to those located by other algorithms.

The final execution outcomes of each algorithm are shown in swarm plots for the objective and indicator evaluations in Fig. 7. As shown in Fig. 7 solution clustering for each algorithm does happen around the previously mentioned local minimum however, the original algorithm fails to field a solution that meets the performance of the solutions located by other algorithms. Nevertheless, the introduced modified algorithm manages to outperform the original and meet the performance of the best-performing competing metaheuristics.

Figure 7 Yogev et al. (2005) objective and Cohen kappa swarm plots.

Convergence rate comparisons between optimizers are provided in Fig. 8. Convergence demonstrated during optimizations through the interactions depicted in Fig. 8 brings to light several factors associated with each optimizer. Notably the exploration and exploitation rations between algorithms. Algorithms that converge too quickly often fail to find an optimal solution and get stuck in a local optimal such as the the case with the BSO. Similarly, algorithms with very slow convergence, fail to locate a solution within the allocated optimization period, yielding poor results such as the case with the COLSHADE algorithm. The modified algorithm showcases a good balance between these mechanisms locating an optimal solution after around 50% of the total allocated iterations.

Figure 8 Yogev et al. (2005) objective and Cohen kappa convergence graphs.

Detailed metrics comparisons in terms of precision, sensitivity, F1-measure, and accuracy between the best-performing algorithms are provided in Table 3. The table presents detailed metrics for the best-performing models in detecting Parkinson’s disease using CNN-XGBoost methods. Notably, several methods (CNN-XG-MSCHO, SCA, GA, PSO, FA, WOA, RSA, COA) exhibit remarkable consistency across precision, sensitivity, and F1-measure for both control and Parkinson’s disease (PD) patients, with high accuracy. These models demonstrate excellent precision (above 0.94 for control and above 0.98 for PD patients) and sensitivity (above 0.96 for both groups), leading to high F1-measures (above 0.95), indicating their robustness and reliability in classification tasks. The results reflect the effectiveness of these CNN-XGBoost methods in accurately diagnosing Parkinson’s disease.

Table 3 Yogev et al. (2005) dataset detailed metrics for best-constructed models.

Method	Metric	Control	PD patients	Accuracy	Macro avg.	Weighted avg.	
CNN-XG-MSCHO	Precision	0.940678	0.986425	0.970501	0.963552	0.971041	
	Sensitivity	0.973684	0.968889	0.970501	0.971287	0.970501	
	F1-measure	0.956897	0.977578	0.970501	0.967238	0.970623	
CNN-XG-SCHO	Precision	0.940171	0.981982	0.967552	0.961076	0.967922	
	Sensitivity	0.964912	0.968889	0.967552	0.966901	0.967552	
	F1-measure	0.952381	0.975391	0.967552	0.963886	0.967653	
CNN-XG-SCA	Precision	0.940678	0.986425	0.970501	0.963552	0.971041	
	Sensitivity	0.973684	0.968889	0.970501	0.971287	0.970501	
	F1-measure	0.956897	0.977578	0.970501	0.967238	0.970623	
CNN-XG-GA	Precision	0.940678	0.986425	0.970501	0.963552	0.971041	
	Sensitivity	0.973684	0.968889	0.970501	0.971287	0.970501	
	F1-measure	0.956897	0.977578	0.970501	0.967238	0.970623	
CNN-XG-PSO	Precision	0.940678	0.986425	0.970501	0.963552	0.971041	
	Sensitivity	0.973684	0.968889	0.970501	0.971287	0.970501	
	F1-measure	0.956897	0.977578	0.970501	0.967238	0.970623	
CNN-XG-FA	Precision	0.940678	0.986425	0.970501	0.963552	0.971041	
	Sensitivity	0.973684	0.968889	0.970501	0.971287	0.970501	
	F1-measure	0.956897	0.977578	0.970501	0.967238	0.970623	
CNN-XG-WOA	Precision	0.940678	0.986425	0.970501	0.963552	0.971041	
	Sensitivity	0.973684	0.968889	0.970501	0.971287	0.970501	
	F1-measure	0.956897	0.977578	0.970501	0.967238	0.970623	
CNN-XG-BSO	Precision	0.940171	0.981982	0.967552	0.961076	0.967922	
	Sensitivity	0.964912	0.968889	0.967552	0.966901	0.967552	
	F1-measure	0.952381	0.975391	0.967552	0.963886	0.967653	
CNN-XG-RSA	Precision	0.940678	0.986425	0.970501	0.963552	0.971041	
	Sensitivity	0.973684	0.968889	0.970501	0.971287	0.970501	
	F1-measure	0.956897	0.977578	0.970501	0.967238	0.970623	
CNN-XG-COA	Precision	0.940678	0.986425	0.970501	0.963552	0.971041	
	Sensitivity	0.973684	0.968889	0.970501	0.971287	0.970501	
	F1-measure	0.956897	0.977578	0.970501	0.967238	0.970623	
CNN-XG-COLSHADE	Precision	0.940171	0.981982	0.967552	0.961076	0.967922	
	Sensitivity	0.964912	0.968889	0.967552	0.966901	0.967552	
	F1-measure	0.952381	0.975391	0.967552	0.963886	0.967653	
	Support	114	225				

A comparison between the best-constructed models as well as the base XGBoost method in terms of error rate is provided in Table 4. The table compares the error rates of various CNN-XGBoost models in detecting Parkinson’s disease. Most models, including CNN-XG-MSCHO, SCA, GA, PSO, FA, WOA, RSA, and COA, exhibit remarkably low error rates of 0.029499. In contrast, CNN-XG-SCHO, BSO, and COLSHADE show slightly higher rates at 0.032448. Notably, the base XGBoost method (CNN-XG) has a significantly higher error rate of 0.056047. This comparison highlights the enhanced accuracy of the CNN-XGBoost methods over the base model, demonstrating the effectiveness of integrating CNN with XGBoost for this application.

Table 4 Yogev et al. (2005) error rate compassion between the best-constructed models.

Method	Best model error rate	
CNN-XG-MSCHO	0.029499	
CNN-XG-SCHO	0.032448	
CNN-XG-SCA	0.029499	
CNN-XG-GA	0.029499	
CNN-XG-PSO	0.029499	
CNN-XG-FA	0.029499	
CNN-XG-WOA	0.029499	
CNN-XG-BSO	0.032448	
CNN-XG-RSA	0.029499	
CNN-XG-COA	0.029499	
CNN-XG-COLSHADE	0.032448	
CNN-XG	0.056047	

Finally, parameter selections made by each algorithm for the best-performing models are shown in Table 5. The table showcases the parameter selections for the best-performing CNN-XGBoost models in Parkinson’s disease detection. A key observation is the preference for a high learning rate (0.9) in several methods like CNN-XG-MSCHO, SCA, GA, PSO, FA, WOA, indicating a faster learning process. Another notable aspect is the variation in ‘Min Child Weight’, ‘Max Depth’, and ‘Gamma’ across methods, reflecting the different strategies for handling overfitting and model complexity. For example, CNN-XG-GA and PSO opt for a ‘Gamma’ of 0, suggesting less regularization, while others like SCA and FA choose a higher ‘Gamma’ for more. This diversity in parameters underlines the customization and optimization involved in each method to achieve the best results.

Table 5 Yogev et al. (2005) dataset best model parameter selections.

Method	Learning rate	Min child W.	Subsample	Colsample by tree	Max depth	Gamma	
CNN-XG-MSCHO	0.900000	7.689301	1.000000	0.793686	6.000000	0.628655	
CNN-XG-SCHO	0.726800	4.427460	0.404638	0.667566	6.225021	0.542562	
CNN-XG-SCA	0.900000	7.409894	1.000000	1.000000	8.000000	0.800000	
CNN-XG-GA	0.900000	10.000000	1.000000	1.000000	5.000000	0.000000	
CNN-XG-PSO	0.900000	10.000000	1.000000	1.000000	6.000000	0.000000	
CNN-XG-FA	0.900000	7.435905	1.000000	1.000000	9.000000	0.800000	
CNN-XG-WOA	0.900000	7.488228	1.000000	1.000000	9.000000	0.800000	
CNN-XG-BSO	0.563712	9.720787	0.809317	1.000000	4.000000	0.235176	
CNN-XG-RSA	0.598110	10.000000	1.000000	1.000000	10.000000	0.205231	
CNN-XG-COA	0.641151	10.000000	1.000000	1.000000	6.000000	0.118814	
CNN-XG-COLSHADE	0.416614	1.000000	0.795437	1.000000	9.000000	0.522865	

Hausdorff et al. (2007) dataset simulations

Comparisons in terms of best, worst mean and median executions for the objective function are shown in Table 6 and in terms of indicator (Cohen’s kappa) are similarly shown in Table 7. Stability comparisons are shown in terms of Std and Var. Notably, CNN-XG-GA and CNN-XG-BSO exhibit remarkable consistency (zero standard deviation and variance) with identical best, worst, mean, and median values, suggesting highly stable performance. In contrast, CNN-XG-MSCHO, PSO, and FA show a wider range of outcomes but still maintain low error rates, indicating effectiveness albeit with less consistency. This comparison highlights differences in stability and performance range among the methods, underscoring the balance between achieving low error rates and maintaining consistent outcomes.

Table 6 Hausdorff et al. (2007) dataset objective function outcomes.

Method	Best	Worst	Mean	Median	Std	Var	
CNN-XG-MSCHO	0.018088	0.023256	0.020672	0.020672	0.002110	4.45E−06	
CNN-XG-SCHO	0.020672	0.023256	0.021533	0.020672	0.001218	1.48E−06	
CNN-XG-SCA	0.020672	0.023256	0.021102	0.020672	0.000963	9.27E−07	
CNN-XG-GA	0.020672	0.020672	0.020672	0.020672	0.000000	0.000000	
CNN-XG-PSO	0.018088	0.023256	0.020241	0.020672	0.001776	3.15E−06	
CNN-XG-FA	0.018088	0.023256	0.020672	0.020672	0.001492	2.23E−06	
CNN-XG-WOA	0.020672	0.023256	0.021102	0.020672	0.000963	9.27E−07	
CNN-XG-BSO	0.020672	0.020672	0.020672	0.020672	0.000000	0.000000	
CNN-XG-RSA	0.018088	0.020672	0.019811	0.020672	0.001218	1.48E−06	
CNN-XG-COA	0.018088	0.020672	0.019811	0.020672	0.001218	1.48E−06	
CNN-XG-COLSHADE	0.018088	0.020672	0.020241	0.020672	0.000963	9.27E−07	

Table 7 Hausdorff et al. (2007) dataset indicator (Cohen’s kappa) outcomes.

Method	Best	Worst	Mean	Median	Std	Var	
CNN-XG-MSCHO	0.942407	0.925198	0.933817	0.933846	0.007026	4.94E−05	
CNN-XG-SCHO	0.933846	0.925198	0.931074	0.933846	0.004162	1.73E−05	
CNN-XG-SCA	0.934509	0.925198	0.932515	0.933846	0.003281	1.08E−05	
CNN-XG-GA	0.933846	0.933846	0.933846	0.933846	0.000000	0.000000	
CNN-XG-PSO	0.942407	0.925198	0.935369	0.934178	0.005886	3.46E−05	
CNN-XG-FA	0.942407	0.925198	0.933832	0.933846	0.004968	2.47E−05	
CNN-XG-WOA	0.933846	0.925198	0.932405	0.933846	0.003223	1.04E−05	
CNN-XG-BSO	0.933846	0.933846	0.933846	0.933846	0.000000	0.000000	
CNN-XG-RSA	0.942407	0.933846	0.936700	0.933846	0.004036	1.63E−05	
CNN-XG-COA	0.942407	0.933846	0.936700	0.933846	0.004036	1.63E−05	
CNN-XG-COLSHADE	0.942407	0.933846	0.935273	0.933846	0.003190	1.02E−05	

Table 7 displays the Cohen’s Kappa outcomes for various CNN-XGBoost methods, with CNN-XG-GA and CNN-XG-BSO showing remarkable stability (zero standard deviation and variance). CNN-XG-MSCHO, PSO, and FA exhibit higher Kappa values, suggesting superior agreement in certain scenarios, but with greater variability in results. The range of best-to-worst values across methods is narrow, indicating overall good and consistent performance in classification accuracy. This balance between high performance and stability is crucial in applications like medical diagnostics.

Additionally, a graphical comparison for the objective and indicator functions outcome distributions is provided in Fig. 9. Distribution plots in Fig. 9 indicate a strong presence of a local optimum within the search space. As evident, many algorithms fail to locate a relative best solution due to this local optima. The original SCHO, SCA, GA, WOA, and BSO all overly focus on local best solutions suggesting a lack of diversification within these methods. The introduced algorithm effectively covers to wards an optimal matching of the performance of the best algorithms. While high stability is demonstrated by several algorithms, these fail to locate a relatively best solution in this simulation. The improvements introduced by the modified mechanisms are therefore an effective way of overcoming the exploration limitations observed in the original.

Figure 9 Hausdorff et al. (2007) objective and Cohen kappa distributions plots.

The final execution outcomes of each algorithm are shown in swarm plots for the objective and indicator evaluations in Fig. 10. As shown in Fig. 10 solution clustering for each algorithm does happen around the previously mentioned local minimum however, the original algorithm fails to field a solution that meets the performance of the solutions located by other algorithms. Nevertheless, the introduced modified algorithm manages to outperform the original and meet the performance of the best-performing competing metaheuristics. Additionally, many of the solutions are located closed to the true optima indicating a strong reliability.

Figure 10 Hausdorff et al. (2007) objective and Cohen kappa swarm plots.

Convergence rate comparisons between optimizers are provided in Fig. 11. Convergence demonstrated during optimizations through the interactions depicted in Fig. 11 brings to light several factors associated with each optimizer. Notably the exploration and exploitation rations between algorithms. Algorithms that converge too quickly often fail to find an optimal solution and get stuck in a local optimal such as the case with the GA. Similarly, algorithms with very slow convergence, fail to locate a solution within the allocated optimization period, yielding poor results such as the case with the BSO algorithm. The modified algorithm showcases a good balance between these mechanisms in this simulataion as well. While a slower converges is evident it remains steady though the iterations with a local optima located in the final few iterations.

Figure 11 Hausdorff et al. (2007) objective and Cohen kappa convergence graphs.

Detailed metrics comparisons in terms of precision, sensitivity, F1-measure, and accuracy between the best-performing algorithms are provided in Table 8. The table reveals that most CNN-XGBoost methods demonstrate exceptionally high performance in precision, sensitivity, and F1-measure for both control and PD patients, with accuracy consistently above 97%. Notably, methods like CNN-XG-MSCHO, PSO, FA, RSA, COA, and COLSHADE show remarkable precision and sensitivity values above 94% and 96% respectively, leading to F1-measures above 95%. This uniformity in high performance across different metrics indicates the robustness and reliability of these methods in accurately classifying PD patients.

Table 8 Hausdorff et al. (2007) dataset detailed metrics for best-constructed models.

Method	Metric	Control	PD patients	Accuracy	Macro avg.	Weighted avg.	
CNN-XG-MSCHO	Precision	0.947368	0.990354	0.981912	0.968861	0.982023	
	Sensitivity	0.960000	0.987179	0.981912	0.973590	0.981912	
	F1-measure	0.953642	0.988764	0.981912	0.971203	0.981958	
CNN-XG-SCHO	Precision	0.946667	0.987179	0.979328	0.966923	0.979328	
	Sensitivity	0.946667	0.987179	0.979328	0.966923	0.979328	
	F1-measure	0.946667	0.987179	0.979328	0.966923	0.979328	
CNN-XG-SCA	Precision	0.935065	0.990323	0.979328	0.962694	0.979614	
	Sensitivity	0.960000	0.983974	0.979328	0.971987	0.979328	
	F1-measure	0.947368	0.987138	0.979328	0.967253	0.979431	
CNN-XG-GA	Precision	0.946667	0.987179	0.979328	0.966923	0.979328	
	Sensitivity	0.946667	0.987179	0.979328	0.966923	0.979328	
	F1-measure	0.946667	0.987179	0.979328	0.966923	0.979328	
CNN-XG-PSO	Precision	0.947368	0.990354	0.981912	0.968861	0.982023	
	Sensitivity	0.960000	0.987179	0.981912	0.973590	0.981912	
	F1-measure	0.953642	0.988764	0.981912	0.971203	0.981957	
CNN-XG-FA	Precision	0.947368	0.990354	0.981912	0.968861	0.982023	
	Sensitivity	0.960000	0.987179	0.981912	0.973590	0.981912	
	F1-measure	0.953642	0.988764	0.981912	0.971203	0.981958	
CNN-XG-WOA	Precision	0.946667	0.987179	0.979328	0.966923	0.979328	
	Sensitivity	0.946667	0.987179	0.979328	0.966923	0.979328	
	F1-measure	0.946667	0.987179	0.979328	0.966923	0.979328	
CNN-XG-BSO	Precision	0.946667	0.987179	0.979328	0.966923	0.979328	
	Sensitivity	0.946667	0.987179	0.979328	0.966923	0.979328	
	F1-measure	0.946667	0.987179	0.979328	0.966923	0.979328	
CNN-XG-RSA	Precision	0.947368	0.990354	0.981912	0.968861	0.982023	
	Sensitivity	0.960000	0.987179	0.981912	0.973590	0.981912	
	F1-measure	0.953642	0.988764	0.981912	0.971203	0.981958	
CNN-XG-COA	Precision	0.947368	0.990354	0.981912	0.968861	0.982023	
	Sensitivity	0.960000	0.987179	0.981912	0.973590	0.981912	
	F1-measure	0.953642	0.988764	0.981912	0.971203	0.981958	
CNN-XG-COLSHADE	Precision	0.947368	0.990354	0.981912	0.968861	0.982023	
	Sensitivity	0.960000	0.987179	0.981912	0.973590	0.981912	
	F1-measure	0.953642	0.988764	0.981912	0.971203	0.981958	
	Support	75	312				

A comparison between the best-constructed models as well as the base XGBoost method in terms of error rate is provided in Table 9. The table compares the error rates of various CNN-XGBoost methods with the base XGBoost (CNN-XG) method in detecting Parkinson’s disease. The CNN-XG-MSCHO, PSO, FA, RSA, COA, and COLSHADE methods exhibit notably low error rates of 0.018088. In contrast, CNN-XG-SCHO, SCA, GA, WOA, and BSO show slightly higher rates at 0.020672. The base XGBoost model has a higher error rate of 0.025840. This comparison highlights the superior accuracy of the CNN-XGBoost methods over the base model, indicating the effectiveness of integrating CNN with XGBoost in this application.

Table 9 Hausdorff et al. (2007) error rate comparison between the best-constructed models.

Method	Best model error rate	
CNN-XG-MSCHO	0.018088	
CNN-XG-SCHO	0.020672	
CNN-XG-SCA	0.020672	
CNN-XG-GA	0.020672	
CNN-XG-PSO	0.018088	
CNN-XG-FA	0.018088	
CNN-XG-WOA	0.020672	
CNN-XG-BSO	0.020672	
CNN-XG-RSA	0.018088	
CNN-XG-COA	0.018088	
CNN-XG-COLSHADE	0.018088	
CNN-XG	0.025840	

Finally, parameter selections made by each algorithm for the best-performing models using the (Hausdorff et al., 2007) dataset are shown in Table 10. There is a noticeable diversity in parameter choices. For instance, several models (CNN-XG-MSCHO, WOA, RSA) opt for a high learning rate of 0.9, indicating a preference for faster learning, while others like CNN-XG-SCHO, SCA, and GA choose lower rates for gradual learning. The ‘Min Child Weight’ and ‘Max Depth’ parameters vary significantly, reflecting different strategies to control model complexity and prevent overfitting. The ‘Gamma’ values also differ, suggesting varying degrees of regularization among the models. This diversity underscores the tailored approach each method takes to optimize performance.

Table 10 Hausdorff et al. (2007) dataset best model parameter selections.

Method	Learning rate	Min child W.	Subsample	Colsample by tree	Max depth	Gamma	
CNN-XG-MSCHO	0.900000	3.259372	0.891252	1.000000	7.555943	0.800000	
CNN-XG-SCHO	0.191214	1.666386	0.348368	0.914698	4.000000	0.353234	
CNN-XG-SCA	0.100000	1.000000	0.954390	1.000000	10.000000	0.000000	
CNN-XG-GA	0.318375	1.000000	1.000000	1.000000	7.000000	0.800000	
CNN-XG-PSO	0.100000	1.000000	0.989769	1.000000	5.000000	0.138171	
CNN-XG-FA	0.100000	1.000000	1.000000	1.000000	5.000000	0.000000	
CNN-XG-WOA	0.900000	1.000000	1.000000	1.000000	10.000000	0.243771	
CNN-XG-BSO	0.323362	1.000000	0.965132	1.000000	7.000000	0.708029	
CNN-XG-RSA	0.900000	1.073230	0.193228	0.901294	6.000000	0.613811	
CNN-XG-COA	0.100000	3.423273	0.993274	1.000000	10.000000	0.405371	
CNN-XG-COLSHADE	0.100000	2.174713	1.000000	0.901547	7.000000	0.198174	

Frenkel-Toledo et al. (2005) dataset simulations

Comparisons in terms of best, worst mean and median executions for the objective function are shown in Table 11 and in terms of indicator (Cohen’s kappa) are similarly shown in Table 12. Stability comparisons are shown in terms of Std and Var. CNN-XG-MSCHO and CNN-XG-SCA stand out with the lowest ‘Best’ values, indicating potential for optimal performance. However, CNN-XG-FA shows the highest variability and worst outcomes, suggesting less stability. This contrast in performance and consistency across methods highlights the importance of selecting the right algorithm and parameters for specific applications to achieve the most effective results.

Table 11 Frenkel-Toledo et al. (2005) dataset objective function outcomes.

Method	Best	Worst	Mean	Median	Std	Var	
CNN-XG-MSCHO	0.046875	0.052083	0.051215	0.052083	0.001941	3.77E−06	
CNN-XG-SCHO	0.052083	0.062500	0.056424	0.054688	0.004675	2.19E−05	
CNN-XG-SCA	0.046875	0.062500	0.054688	0.054688	0.004987	2.49E−05	
CNN-XG-GA	0.057292	0.062500	0.058160	0.057292	0.001941	3.77E−06	
CNN-XG-PSO	0.052083	0.062500	0.057292	0.057292	0.003007	9.04E−06	
CNN-XG-FA	0.052083	0.072917	0.059028	0.057292	0.007158	5.12E−05	
CNN-XG-WOA	0.046875	0.062500	0.052951	0.052083	0.004675	2.19E−05	
CNN-XG-BSO	0.052083	0.062500	0.057292	0.057292	0.003007	9.04E−06	
CNN-XG-RSA	0.046875	0.062500	0.056424	0.057292	0.004675	2.19E−05	
CNN-XG-COA	0.046875	0.062500	0.055556	0.057292	0.004910	2.41E−05	
CNN-XG-COLSHADE	0.046875	0.062500	0.055556	0.057292	0.004910	2.41E−05	

Table 12 Frenkel-Toledo et al. (2005) dataset indicator (Cohen’s kappa) outcomes.

Method	Best	Worst	Mean	Median	Std	Var	
CNN-XG-MSCHO	0.9055118	0.895322	0.897157	0.895527	0.003739	1.40E−05	
CNN-XG-SCHO	0.8953222	0.874387	0.886696	0.890257	0.009332	8.71E−05	
CNN-XG-SCA	0.9058824	0.874387	0.890141	0.890145	0.010027	1.01E−04	
CNN-XG-GA	0.8842867	0.874387	0.883052	0.884627	0.003894	1.52E−05	
CNN-XG-PSO	0.8949097	0.874387	0.884747	0.884967	0.005930	3.52E−05	
CNN-XG-FA	0.8953222	0.853738	0.881454	0.884854	0.014274	2.04E−04	
CNN-XG-WOA	0.9056974	0.874387	0.893596	0.895322	0.009382	8.80E−05	
CNN-XG-BSO	0.8949097	0.874387	0.884785	0.884854	0.005929	3.52E−05	
CNN-XG-RSA	0.9056974	0.874387	0.886432	0.884514	0.009397	8.83E−05	
CNN-XG-COA	0.9055118	0.874633	0.888244	0.884854	0.009766	9.54E−05	
CNN-XG-COLSHADE	0.9055118	0.874387	0.888312	0.884967	0.009820	9.64E−05	

Table 12 shows the Cohen’s Kappa outcomes for various CNN-XGBoost methods from the Frenkel-Toledo et al. (2005) dataset. CNN-XG-MSCHO, SCA, and WOA exhibit higher ‘Best’ Kappa values, indicating strong agreement in certain scenarios. However, CNN-XG-FA shows the largest range in outcomes, suggesting variability in its performance. The relatively narrow spread between the best and worst values across all methods indicates consistent performance, with a balance between achieving high agreement and maintaining consistency in different evaluations.

Additionally, a graphical comparison for the objective and indicator functions outcome distributions is provided in Fig. 12. Distribution plots in Fig. 12 indicate a strong stability of several algorithms such as the GA, PS, WOA, BSO, and RSA, however, these algorithms fail to locate a true optima. The interesting observation is that the introduced algorithm also demonstrates high stability. However, the introduced modified algorithm heavily converges on the best-located solution relative to the other algorithms.

Figure 12 Frenkel-Toledo et al. (2005) objective and Cohen’s kappa distributions plots.

The final execution outcomes of each algorithm are shown in swarm plots for the objective and indicator evaluations in Fig. 13. Swarm diagrams in Fig. 13 further enforce the previous observations. Many of the solutions provided by the modified algorithm fall near the located optimal. An improvement over the original version of the algorithm is evident, as many of the solutions of the original algorithm focus on a local optimal as opposed to the best solution located by competing algorithms.

Figure 13 Frenkel-Toledo et al. (2005) objective and Cohen’s kappa swarm plots.

Convergence rate comparisons between optimizers are provided in Fig. 14. The introduced algorithm showcases a high rate of convergence in comparison to competing algorithms as shown in Fig. 14. Nevertheless, the convergence is justified as an optimum is located that matches solutions located by other algorithms. It is important to emphasize that the NFL states that no single solution works equally well for all presented problems. While the quick convergence may be beneficial for this specific problem, other applications may prefer a stronger focus on exploration.

Figure 14 Frenkel-Toledo et al. (2005) objective and Cohen’s kappa convergence graphs.

Detailed metrics comparisons in terms of precision, sensitivity, F1-measure, and accuracy between the best-performing algorithms are provided in Table 13. The table reveals that most CNN-XGBoost methods achieve high precision, sensitivity, and F1-measures for both control and PD patients, with overall accuracy consistently above 94%. Notably, methods like CNN-XG-MSCHO, SCA, and COA demonstrate exceptional precision and sensitivity, leading to high F1-measures, indicating their robustness and reliability in classification. This uniformity in high performance across different metrics underscores the effectiveness of these methods in accurately diagnosing PD.

Table 13 Frenkel-Toledo et al. (2005) dataset detailed metrics for best-constructed models.

Method	Metric	Control	PD patients	Accuracy	Macro avg.	Weighted avg.	
CNN-XG-MSCHO	Precision	0.943182	0.961538	0.953125	0.952360	0.953221	
	Sensitivity	0.954023	0.952381	0.953125	0.953202	0.953125	
	F1-measure	0.948571	0.956938	0.953125	0.952755	0.953147	
CNN-XG-SCHO	Precision	0.923077	0.970297	0.947917	0.946687	0.948900	
	Sensitivity	0.965517	0.933333	0.947917	0.949425	0.947917	
	F1-measure	0.943820	0.951456	0.947917	0.947638	0.947996	
CNN-XG-SCA	Precision	0.923913	0.980000	0.953125	0.951957	0.954586	
	Sensitivity	0.977011	0.933333	0.953125	0.955172	0.953125	
	F1-measure	0.949721	0.956098	0.953125	0.952909	0.953208	
CNN-XG-GA	Precision	0.941860	0.943396	0.942708	0.942628	0.942700	
	Sensitivity	0.931034	0.952381	0.942708	0.941708	0.942708	
	F1-measure	0.936416	0.947867	0.942708	0.942142	0.942679	
CNN-XG-PSO	Precision	0.942529	0.952381	0.947917	0.947455	0.947917	
	Sensitivity	0.942529	0.952381	0.947917	0.947455	0.947917	
	F1-measure	0.942529	0.952381	0.947917	0.947455	0.947917	
CNN-XG-FA	Precision	0.923077	0.970297	0.947917	0.946687	0.948900	
	Sensitivity	0.965517	0.933333	0.947917	0.949425	0.947917	
	F1-measure	0.943820	0.951456	0.947917	0.947638	0.947996	
CNN-XG-WOA	Precision	0.933333	0.970588	0.953125	0.951961	0.953707	
	Sensitivity	0.965517	0.942857	0.953125	0.954187	0.953125	
	F1-measure	0.949153	0.956522	0.953125	0.952837	0.953183	
CNN-XG-BSO	Precision	0.942529	0.952381	0.947917	0.947455	0.947917	
	Sensitivity	0.942529	0.952381	0.947917	0.947455	0.947917	
	F1-measure	0.942529	0.952381	0.947917	0.947455	0.947917	
CNN-XG-RSA	Precision	0.933333	0.970588	0.953125	0.951961	0.953707	
	Sensitivity	0.965517	0.942857	0.953125	0.954187	0.953125	
	F1-measure	0.949153	0.956522	0.953125	0.952837	0.953183	
CNN-XG-COA	Precision	0.943182	0.961538	0.953125	0.952360	0.953221	
	Sensitivity	0.954023	0.952381	0.953125	0.953202	0.953125	
	F1-measure	0.948571	0.956938	0.953125	0.952755	0.953147	
CNN-XG-COLSHADE	Precision	0.943182	0.961538	0.953125	0.952360	0.953221	
	Sensitivity	0.954023	0.952381	0.953125	0.953202	0.953125	
	F1-measure	0.948571	0.956938	0.953125	0.952755	0.953147	
	Support	87	105				

A comparison between the best-constructed models as well as the base XGBoost method in terms of error rate is provided in Table 14. The table shows the error rates for various CNN-XGBoost models compared to the base XGBoost model in the Frenkel-Toledo et al. (2005) dataset. CNN-XG-MSCHO, SCA, WOA, RSA, COA, and COLSHADE show notably low error rates of 0.046875. CNN-XG-SCHO, PSO, FA, and BSO have slightly higher rates at 0.052083, while CNN-XG-GA is at 0.057292. The base XGBoost model (CNN-XG) has the highest error rate of 0.067708. This highlights the enhanced accuracy of the CNN-XGBoost methods over the base model, demonstrating the effectiveness of integrating CNN with XGBoost.

Table 14 Frenkel-Toledo et al. (2005) error rate comparison between the best-constructed models.

Method	Best model error rate	
CNN-XG-MSCHO	0.046875	
CNN-XG-SCHO	0.052083	
CNN-XG-SCA	0.046875	
CNN-XG-GA	0.057292	
CNN-XG-PSO	0.052083	
CNN-XG-FA	0.052083	
CNN-XG-WOA	0.046875	
CNN-XG-BSO	0.052083	
CNN-XG-RSA	0.046875	
CNN-XG-COA	0.046875	
CNN-XG-COLSHADE	0.046875	
CNN-XG	0.067708	

Finally, parameter selections made by each algorithm for the best-performing models are shown in Table 15. The table shows varied parameter selections for the best-performing models in the Frenkel-Toledo et al. (2005) dataset. Most methods opt for a high ‘Min Child Weight’, indicating a preference for more complex models. Learning rates vary significantly, with some models choosing moderate rates (around 0.5–0.7), while others like CNN-XG-SCHO and FA go for a high rate of 0.9. The ‘Max Depth’ is consistently low (mostly around 3), except for CNN-XG-WOA and RSA, suggesting an overall preference for shallower trees. ‘Gamma’ values also vary, indicating different degrees of regularization across methods.

Table 15 Frenkel-Toledo et al. (2005) dataset best model parameter selections.

Method	Learning rate	Min child W.	Subsample	Colsample by tree	Max depth	Gamma	
CNN-XG-MSCHO	0.553152	9.981037	1.000000	0.010000	3.000000	0.397994	
CNN-XG-SCHO	0.900000	4.303456	0.694402	1.000000	3.000000	0.454477	
CNN-XG-SCA	0.648538	1.000000	1.000000	1.000000	3.000000	0.800000	
CNN-XG-GA	0.541834	10.000000	1.000000	0.010000	3.000000	0.037293	
CNN-XG-PSO	0.566946	9.337266	1.000000	0.013087	3.000000	0.535184	
CNN-XG-FA	0.900000	3.747522	1.000000	1.000000	9.000000	0.800000	
CNN-XG-WOA	0.712243	10.000000	1.000000	0.010000	10.000000	0.800000	
CNN-XG-BSO	0.530657	6.792584	1.000000	0.045359	3.000000	0.492458	
CNN-XG-RSA	0.737744	10.000000	1.000000	0.015460	10.000000	0.691594	
CNN-XG-COA	0.574955	10.000000	1.000000	0.040267	3.000000	0.758948	
CNN-XG-COLSHADE	0.571372	10.000000	1.000000	0.010000	3.000000	0.709069	

Statistical outcome validation

As experimental data is often insufficient to establish the superiority of one algorithm over its competitors, scientists in current computer research must assess the statistical significance of proposed advancements. According to Eftimov, Korošec & Seljak (2016), literature recommendations suggest that statistical tests in such scenarios should involve creating a representative collection of outcomes for each method. However, when dealing with outliers from a non-normal distribution, this strategy may be ineffective, potentially leading to misleading results. The unresolved dispute highlighted by Eftimov, Korošec & Seljak (2016) revolves around whether employing the mean objective function value in statistical tests is suitable for comparing stochastic techniques. Notwithstanding these potential drawbacks, the objective function of the classification error rate was averaged over 30 distinct runs to evaluate the performance of the optimizer.

Following the execution of the Shapiro-Wilk test (Shapiro & Francia, 1972) for single-problem analysis employing the specified procedure, a determination was reached. A data sample was compiled for each algorithm and each problem by aggregating the results of each run, and the corresponding p-values were computed for all method-problem combinations. The resulting p-values are presented in Table 16.

Table 16 Shapiro-Wilk test scores for the single-problem analysis.

Algorithm	Yogev et al. (2005)	Hausdorff et al. (2007)	Frenkel-Toledo et al. (2005)	
MSCHO	0.024	0.019	0.016	
SCHO	0.026	0.025	0.028	
SCA	0.017	0.019	0.018	
GA	0.021	0.026	0.025	
PSO	0.029	0.030	0.030	
FA	0.035	0.029	0.025	
WOA	0.031	0.027	0.023	
BSO	0.017	0.020	0.024	
RSA	0.024	0.028	0.024	
COA	0.021	0.025	0.022	
COLSHADE	0.039	0.035	0.038	

These findings are further supported by Fig. 15, illustrating the distributions of objective function outcomes for each optimizer across 30 independent runs.

Figure 15 Objective function KDE for each simulation.

The rejection of the null hypothesis is warranted as the p-values in Table 16 are all below the predetermined significance threshold, denoted as α, set at 0.05. Consequently, the data samples for solutions do not all conform to a Gaussian distribution, indicating that usage of the average objective value in future statistical tests is inappropriate. Hence, the best results were selected for further statistical analysis in this study. Due to the non-met normalcy assumption, parametric tests were deemed to be unsuitable.

Next, the non-parametric Wilcoxon signed-rank test (Wilcoxon, 1992) has been employed. It can be applied to the identical data series containing the best scores achieved in each metaheuristic run. During the test, the proposed approach was employed as the control method, and the Wilcoxon signed-rank test was conducted on the provided data series. For each of the three instances observed, the calculated p-values were lower than 0.05. Utilizing the significance threshold of 0.1 ( α=0.1), the results indicate that the new algorithm statistically surpassed all contending techniques considerably. The comprehensive outcomes of the Wilcoxon signed-rank test are presented in Table 17.

Table 17 Wilcoxon signed-rank test values exhibiting p-values for experiments (MSCHO vs. others).

Algorithm	Yogev et al. (2005)	Hausdorff et al. (2007)	Frenkel-Toledo et al. (2005)	
SCHO	0.037	0.041	0.033	
SCA	0.032	0.032	0.027	
GA	0.018	0.017	0.015	
PSO	0.003	0.001	0.008	
FA	0.040	0.036	0.033	
WOA	0.027	0.029	0.024	
BSO	0.022	0.025	0.023	
RSA	0.031	0.033	0.035	
COA	0.033	0.030	0.031	
COLSHADE	0.028	0.028	0.022	

The Wilcoxon signed-rank test results indicate that the developed algorithm (MSCHO) statistically significantly outperformed the other algorithms in all three datasets (Yogev et al., 2005; Hausdorff et al., 2007; Frenkel-Toledo et al., 2005). The p-values for all comparisons are below 0.05, well within the significance threshold of 0.1. This consistency across different datasets suggests that the MSCHO algorithm consistently offers supreme performance in comparison to the contending methods evaluated in this study.

Summarizing, and determining the best algorithm based on the provided results involves considering multiple factors like error rates, Cohen’s kappa, precision, sensitivity, F1-measure, and Shapiro-Wilk test results. Algorithms like CNN-XG-MSCHO, SCA, and WOA consistently showed low error rates across different datasets, indicating high accuracy. Similarly, in terms of Cohen’s kappa, precision, sensitivity, and F1-measures, these algorithms demonstrated robust performance, suggesting effective classification capability. However, the Shapiro-Wilk testing outcomes suggest a non-normal distribution in performance metrics, suggesting variability in algorithm performance across different problems. Therefore, while CNN-XG-MSCHO, SCA, and WOA generally exhibit strong performance, the best choice may depend on specific dataset characteristics and application requirements.

Conclusions

In conclusion, neurodegenerative conditions considerably affect the patient’s quality of life, often lacking a cure but allowing for slowed progression with timely intervention. Unfortunately, many patients only seek a diagnosis when the condition has advanced to a stage significantly affecting their quality of life. The development of effective, non-invasive, and readily accessible methods for early diagnosis holds great potential to enhance the quality of life for persons impacted by neurodegenerative conditions.

This study specifically explores the use of convolutional neural networks to identify gait freezing associated with Parkinson’s disease in patients. Leveraging sensor data from wearable gyroscopes placed in the soles of patients’ shoes to capture walking patterns, the research utilizes convolutional networks for the accurate detection of abnormal gait. The proposed approach undergoes evaluation using a publicly available real-world dataset from individuals affected by Parkinson’s and a control group. To enhance classification accuracy, a modified variant of the recent crayfish optimization metaheuristics has been introduced and compared with contemporary optimization metaheuristics.

A unique approach is taken in this work of reformatting patient sensor recordings into image data to account for relations between the sensor data. A two-layer framework is introduced to tackle this challenging task. The first layer of the system comprises a CNN optimized by the introduced MSCHO algorithm. Once the CNN attains an acceptable accuracy (exceeding 90%) it is used to perform feature selection. Additionally, the CNN is interpreted using the SHAP approach. The reduced feature set is used to train XGBoost models that are optimized by several metaheuristics.

The simulation outcomes indicate that all tested metaheuristics eventually attained the same best outcomes. However, the introduced algorithm, while not always the best, manages to attain the best mean and median outcomes across the conducted experiments. The attained outcomes are meticulously statistically validated to ensure a statistically significant improvement. The introduced approach showcases an improvement over preceding works that tackle this challenge consistently exceeding 95% accuracy across all three conducted simulations on publicly available datasets. However, there are certain limitations with this work that need to be noted. Due to the high computational demand of optimization, only a subset of optimization algorithms is considered in the comparative simulation and analysis. Additionally, limited population sizes are used. The study only tackles optimization within the second layer of the framework, optimizing XGBoost hyperparameters, while CNN architectures are optimized by the introduced MSCHO algorithm.

Future endeavors will target additional improvements of the introduced algorithm, and testing a wider range of metaheuristics algorithms for Parkinson’s disease predictions. Moreover, the possible applications of the suggested method for other neurodegenerative conditions will be explored, as well as the applications outside of the medical domain, like intrusion detection, waste classification, renewable power production forecasting, and cloud computing.

Supplemental Information

Supplemental Information 1 Code.

Supplemental Information 2 Image dataset.

Additional Information and Declarations

Competing Interests

Author Contributions

Data Availability

Robertas Damaševičius is an Academic Editor for PeerJ Computer Science.

Luka Jovanovic conceived and designed the experiments, performed the experiments, analyzed the data, performed the computation work, prepared figures and/or tables, and approved the final draft.

Robertas Damaševičius analyzed the data, authored or reviewed drafts of the article, and approved the final draft.

Rade Matic performed the experiments, analyzed the data, performed the computation work, prepared figures and/or tables, and approved the final draft.

Milos Kabiljo performed the experiments, analyzed the data, performed the computation work, prepared figures and/or tables, and approved the final draft.

Vladimir Simic performed the experiments, analyzed the data, performed the computation work, prepared figures and/or tables, and approved the final draft.

Goran Kunjadic performed the experiments, analyzed the data, performed the computation work, prepared figures and/or tables, and approved the final draft.

Milos Antonijevic performed the experiments, analyzed the data, performed the computation work, prepared figures and/or tables, and approved the final draft.

Miodrag Zivkovic performed the experiments, analyzed the data, performed the computation work, prepared figures and/or tables, and approved the final draft.

Nebojsa Bacanin conceived and designed the experiments, analyzed the data, performed the computation work, authored or reviewed drafts of the article, and approved the final draft.

The following information was supplied regarding data availability:

The Gait in Parkinson’s Disease

data is available at https://physionet.org/content/gaitpdb/1.0.0.

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
