# Peer review of "Detecting Parkinson’s disease from shoe-mounted accelerometer sensors using convolutional neural networks optimized with modified metaheuristics"

_PeerJ Computer Science, doi:10.7717/peerj-cs.2031_

## Round 0.1 · original submission · Major Revisions

Dear authors,

Thank you for your submission. Your article has not been recommended for publication in its current form. However, we do encourage you to address the concerns and criticisms of the reviewers, especially about experiment design and results and resubmit your article once you have updated it accordingly.

Best wishes,

**Language Note:** PeerJ staff have identified that the English language needs to be improved. When you prepare your next revision, please either (i) have a colleague who is proficient in English and familiar with the subject matter review your manuscript, or (ii) contact a professional editing service to review your manuscript. PeerJ can provide language editing services - you can contact us at [email protected] for pricing (be sure to provide your manuscript number and title). – PeerJ Staff

Reviewer 1 ·

Basic reporting

1-The paper, generally states that the CNN and Metaheuristic-based method developed for the early detection of neurodegenerative diseases achieves higher accuracy among existing methods. It is stated that early diagnosis of the disease is possible by classifying the data obtained from some sensors placed under the shoes. Although the study is exciting in this respect, it is a fact that there are some parts of the article that need correction.
2- It seems that unnecessary abbreviations are used. To use an abbreviation, the abbreviated expression must appear in at least two places in the article. Abbreviations such as "Amyotrophic Lateral Sclerosis (ALS), Multiple Sclerosis (MS), Creutzfeldt-Jakob Disease (CJD)" should be removed.
3-Although it is clear that the study is for the detection of Parkinson's disease, it is seen that Alzheimer's disease and neurodegenerative diseases are included too much in the introduction part of the article and excessive references are given to related studies. This part should be reduced slightly.
4-It is seen that only a literature review is done and scientific studies are not approached critically. The literature should be examined critically.
5-It is thought that not enough studies have been done on CNN. Studies such as Classification of brain tumor images using deep learning methods should be examined.
6-Is the main gap in the literature that the signals obtained from the sensors placed under the shoes of Parkinson's patients are converted into images, fed to CNN architectures, and then classified? The fundamental contribution is not fully understood.

Experimental design

7- Metaheuristic optimization methods have a total of 9 types in terms of the natural phenomenon they are inspired by: light, swarm, physics, biology, chemistry, music, sports, mathematics, and hybrid-based. Authors need to cite light-based optimization algorithms in the literature and rearrange this section.
8-The authors used the sinh cosh optimizer algorithm in the experiments. Why didn't the authors use the Sine Cosine Algorithm?
9-Figure 1 is very difficult to understand and needs to be improved.
10-The authors stated in the article, "An additional two columns are conducted in the dataset that represents the sum of all the sensors for each foot. This is done to account for variance in patient weight and variance in balance." Although this part is very important, it is not fully understood why the two legs are combined. In addition, it is said that the data from these sensors are collected, should we understand the juxtaposition of the outputs of the sensors? This part is not clear and should be reconsidered.
11-It is mentioned that after converting the data from the sensors into a 2D image, the image is lengthened by improvement. This part should be explained. How is the stretching of the image accomplished?

Validity of the findings

12-In the article, the authors say "The population size of 10 agents is used by each algorithm with 15 iterations allocated to improve outcomes. To account for the randomness inherent in metaheuristics algorithms simulations are carried out through 30 experiments. In addition to optimized models, a baseline XGBoost implementation without optimization is included in the comparison."

I believe that 15 iterations is quite low for metaheuristic algorithms. I request that the experiments be repeated with 50 iterations and the results be added to the article. Thus, it will be revealed how much the increase in the number of iterations affects the accuracy.

·

Basic reporting

Clear Problem Statement: Emphasizes the negative impact of delayed diagnosis on patient quality of life. Identifies the need for non-invasive and accessible early detection methods.

Experimental design

Proposed Solution: Describes the application of CNNs for analyzing gait patterns from wearable sensor data to detect abnormal walking patterns associated with Parkinson's disease.

SHAP Explanation: How does it add value to the overall approach?

Validity of the findings

Evaluation and Results:
Mentions the use of a publicly available dataset and comparison with a control group.
Introduces a modified optimization algorithm (MSCHO) and claims superior accuracy compared to existing methods.

Metrics Details:
While mentioning low error rates and high scores (Cohen's Kappa, precision, recall, F1-score), providing specific values would strengthen the claim.
Briefly elaborate on the significance of these metrics in evaluating the model's performance.

Generalizability and Limitations:
Discuss potential limitations of the approach, such as data dependence or real-world implementation challenges.
If applicable, briefly mention the generalizability of the findings to other neurodegenerative conditions.

MSCHO Algorithm:
How does it differ from the original algorithm and how does it contribute to improved performance?

Additional comments

Impact and Conclusion:
Emphasizes the potential for early, non-invasive diagnosis using the proposed approach, ultimately leading to improved patient outcomes.

Consider mentioning potential future work directions, such as expanding the algorithm comparison or exploring applications for other conditions. If applicable, you can briefly touch upon the ethical considerations of using wearable sensor data for medical diagnosis.

---

## Round 0.2 · accepted · Accept

Dear authors,

Thank you for the revision and for clearly addressing all the reviewers' comments. I confirm that the paper is improved and addresses the reviewers' concerns. Your paper is now acceptable for publication in light of this revision.

Best wishes,

Reviewer 1 ·

Basic reporting

The authors have accomplished everything that was asked of them.

Experimental design

The authors have accomplished everything that was asked of them.

Validity of the findings

The authors have accomplished everything that was asked of them.

Additional comments

The authors have accomplished everything that was asked of them.